# Enhancement of the Water-Lubricated Tribological Properties of Hybrid PTFE/Nomex Fabric Laminate Composite via Epoxy Resin and Graphite Filler

**DOI:** 10.3390/ma15010062

**Published:** 2021-12-22

**Authors:** Ying Liu, Gengyuan Gao, Dan Jiang, Zhongwei Yin

**Affiliations:** School of Mechanical Engineering, Shanghai Jiao Tong University, Shanghai 200240, China; jiangdan@sjtu.edu.cn (D.J.); yinzw1972@163.com (Z.Y.)

**Keywords:** fabric composite, epoxy, graphite, friction and wear, water-lubricated

## Abstract

This paper studied a hybrid polytetrafluoroethylene (PTFE)/Nomex fabric composite with lower friction coefficient (COF) and high underwater wear resistance. A pin-on-disk tribometer was used to test tribological properties under different applied loads and rotation speeds. The wear surface, transfer film and cross-section were analyzed by scanning electron microscope (SEM) and optical microscope. The results showed enhanced underwater tribological properties because of excellent self-lubricating properties of PTFE fibers and a good lubricating effect and load-carrying capacity of graphite fillers. Improved underwater mechanical strength was connected to the high strength of epoxy resin and high bonding force between Nomex and epoxy resin.

## 1. Introduction

Hybrid PTFE/Nomex fabric resin laminate composite has high mechanical and tribological properties that can be used in aviation, aerospace, shipbuilding and so on. In the shipbuilding field, a fabric resin laminate composite can be used as water-lubricated stern tube bearing materials of ships using water as lubricant [1,2,3,4]. PTFE fibers provide excellent lubricating effects, and Nomex fibers and bonding resin are tightly combined to provide high strength of laminate composite.

Epoxy resin is an excellent adhesion resin [5,6,7,8]. Zahabi et al. [9] investigated the tribological performance of PTFE/Glass fabric/epoxy resin composite. The results showed good self-lubricating features and high tribological performance. Larsen et al. [10] studied the tribological properties of epoxy resin by adding PTFE microparticles. The results showed an excellent synergy between the epoxy resin and PTFE in terms of improving friction and reducing wear.

Meanwhile, phenolic resin is also widely used as adhesive resin for hybrid PTFE/Nomex fabric composite [11]. However, PTFE fibers and phenolic resin have weak adhesion causing low bonding force between fabric layers [12]. Phenolic resin mainly infiltrates Nomex fibers and solidifies it. As a result, the diameter of PTFE and Nomex fibers has a significant effect on working surface roughness. Phenolic resin and epoxy resin are both excellent adhesives. Epoxy resin has stronger bonding forces and better synergistic effects with PTFE fibers [13,14]. In this paper, epoxy resin is used instead of phenolic resin as the adhesive resin.

Adding lubricating fillers to adhesive resin is beneficial for the improvement of the tribological properties. Many studies are devoted to various fillers (graphite, carbon fabrics, ZrB_2_, TiO_2_, SiO_2_, Mo_2_C, CuS and MoS_2_) of the hybrid PTFE/Nomex fabric composite under dry sliding conditions [15,16,17,18,19,20]. Among them, graphite fillers had good comprehensive properties and were commonly used [21,22]. Zhang and Du [23] investigated epoxy-resin graphene composites, and the results showed that, when graphite content reaches 5%, the exfoliated graphene was most easily bonded to the metal pin to form a transfer film, reducing wear most effectively. Zhang et al. [24] investigated the effect of graphite on the tribological behaviors of hybrid PTFE/Nomex fabric composite. The results showed that graphite filler could significantly improve wear resistance because graphite improved the thermal stability of the composite. Ren and Wang [25] investigated the effect of graphite/graphene on the tribological behaviors of PTFE/Nomex fabric-reinforced phenolic composite. The results showed that graphite/graphene fillers had apparent effects on friction reduction and wear resistance of composite materials. The rolling effect of graphene reduced COF, and the high load-carrying of graphite reduced wear. Therefore, in this paper, a graphite filler is used in the fabric laminate composite.

The working surface has a good modification effect under dry sliding; however, tribological properties are reduced under water lubrication [26]. Ren et al. [27] investigated the tribological properties of hybrid PTFE/Nomex fabric/phenolic composite underwater, and the results showed that the working surface was severely damaged, and the phenolic resin peeled off under distilled water and seawater; compared to seawater, the fabric composite wears more seriously under pure water lubrication. Meng et al. [28] investigated friction and wear behavior of polyamide 6 composites under dry sliding and water-lubricated conditions. They found that the material exhibited a higher wear rate underwater than dry friction. Due to the fact that water absorption reduced the strength of the material, it is not conducive to the formation of transfer film underwater. Chen et al. [29] investigated the tribological properties of phenolic composites by adding carbon fabric under pure water. The results showed that a decrease in interlaminar shear strength and increase in wear rate were most obvious under pure water compared to dry friction and under seawater. Therefore, we used pure water as the lubricant in this paper.

In order to improve tribological properties underwater, some methods are proposed in the literature. Hu et al. [30] studied hydrophilic PVA fiber-reinforced thermoplastic polyurethane materials for water-lubricated stern bearing. The results revealed that the addition of hydrophilic PVA fibers could improve the tribological properties of thermoplastic polyurethane (TPU) because hydrophilic PVA fibers improved the affinity and surface storage ability relative to water. Liu et al. [31] researched carbon nanotube (CNT) enhancement on the tribological performance of high-strength glass fabric/phenolic laminate underwater. The results showed that the friction coefficient of the laminate was remarkably stabilized and lowered underwater by the incorporation of CNTs. Due to the interface of high-strength glass fabric/phenolic, the interlaminar shear strength of the laminate was improved by the incorporation of CNTs.

The above methods used various fillers to improve underwater tribological properties. However, the strengths of phenolic resin and Nomex fibers decreased in water, and the decreasing bonding force between phenolic resin and Nomex fibers caused low adhesion interlayers. Moreover, fillers improving the effect of adhesion between layers are limited. Phenolic resin has difficulty providing enough strength underwater. Consequently, phenolic resin was more likely to be worn away; after that, the Nomex fibers lost the resin package and were worn faster, causing large COF and high wear amount. Therefore, a high strength and stable performance adhesive resin was needed to improve the underwater tribological properties of hybrid PTFE/Nomex fabric composite. Unfortunately, there are few studies on enhancing the underwater tribological properties of hybrid PTFE/Nomex fabric laminate composite. In this paper, a more suitable epoxy resin was used under water lubrication. With the assistance of graphite filler, the hybrid fabric laminate composite has excellent tribological properties both under dry sliding and water lubrication. The effect of epoxy resin, hybrid PTFE/Nomex fabric and graphite filler was analyzed under water lubrication.

## 2. Experiment

### 2.1. Equipment and Sample Preparation

PTFE (density: 2.2 g/cm^3^; elongation: 50%) and Nomex (density: 1.36 g/cm^3^; elongation: 32%) fibers were produced by DuPont, Wilmington, DE, USA). The resin adopts thermosetting epoxy resin (Shanghai Xinguang Chemical Co., Ltd., Shanghai, China) that is commercially available. Figure 1 showed the preparation process of hybrid fabric samples. In Figure 1a,b, PTFE and Nomex fibers were woven into hybrid PTFE/Nomex fabrics on the weaving machine (SXACT-C, China). The microstructure of the fabric is shown in Figure 1c, the working surface was rich in PTFE fibers with 75% and the bonding surface was rich in Nomex fibers with 75%. Then, graphite (Shanghai Aladdin Biochemical Technology Co., Ltd., Shanghai, China) (grain size: 13 µm; density: 1.04 g/m^3^; specific surface area: 2.63 m^2^/g) was added to the epoxy resin, as shown in Figure 1d,e. Then, they were stirred well using ultrasonic methods. Later, the resin was applied to the hybrid PTFE/Nomex fabrics squares as even as possible, weighed and the relative mass fraction of the resin was calculated after drying for 2 h at 80 °C. Immersion was repeated several times until the relative mass fraction of the resin reached 40 ± 5%. After that, the pre-impregnated fabrics were placed together one by one for a total of 20 layers, as shown in Figure 1f. A curing press was used to consolidate pre-impregnated fabrics under 150 °C at 3 MPa for 2 h. as shown in Figure 1g. Finally, the samples were taken out and cut into 40 mm x 40 mm to make the test fabric samples shown in Figure 1h.

### 2.2. Friction and Wear Test

Figure 2A shows the schematics of a pin-on-disk tribometer (RTEC MFT-500, USA), and a stationary steel pin slides against a rotating test sample, which was fixed on a rotational steel disk. Figure 2B shows the photographs under dry sliding. The water lubrication method consisted of dripping distilled water onto the fabric sample at a rate of 60 drops per minute as shown in Figure 2C. The fabric sample was soaked in pure water for 100 h before the water lubrication test. A flat-ended GCr15 pin (diameter 4 mm) was secured to the load arm with a chuck. The pin was polished with 800-grade waterproof abrasive papers before using. The distance between the center of the pin and the center of the axis was 18 mm. Friction and wear tests were performed under laboratory conditions (temperature, 25 °C; relative humidity, ~50%). The rotational speeds were 100, 200, 300, 400 and 500 rpm, respectively; the loads were 1, 2, 3, 4 and 5 MPa at every sliding speed. Every test lasted for 10 min and was repeated three times, using the average value as the test result.

The friction coefficient (COF) was measured from the frictional torque gained by a load cell sensor and could be directly read from the computer running the friction measurement software (RTEC, USA). The worn surfaces were analyzed using scanning electron microscopy (SEM) and optical microscope. The 3D profile, surface roughness (Sa) and wear volume (v1, v2) of the working surface was measured via a laser microscope, as shown in Figure 2D.

## 3. Results and Discussion

### 3.1. The Results of Friction Coefficient (COF)

Figure 3 shows the COF of fabric sample under dry sliding. In Figure 3a,b, the higher the speed and the lower the load, the greater COF fluctuation was. In Figure 3c,d the greater the load, the smaller the average COF. Load has a larger effect on average COF than speed under dry sliding, and increasing the load resulted in lower dry sliding COF. Due to the fact that a large load improves the actual force area between friction pairs, it decreases contact pressure. Moreover, large load makes it easier to form a PTFE film on the fabric sample surface and metal pin. High speed increases the fluctuation of COF, the fabric sample surface roughness is worn out more quickly with the production of abrasive particles. Under the secondary lubrication of abrasive particles, COF is reduced with speed increasing. Ultimately, the dry sliding COF is lower under great load and high speed [32].

Figure 4 shows COF under water lubrication after soaking. In Figure 4a,b, the trends of COF with time are not similar under different loads and speeds. Due to the fact that the water lubricating state influences the trends of COF, different loads and speeds result in different lubricating states. Figure 4c shows the average COF gradually decreasing with speed increasing; in Figure 4d, the average COF first reduces and then increases with load addition reaching a lower COF at 2–3 MPa. With respect to underwater lubrication, because of the high strength and excellent adhesive of epoxy resin, fabric sample hardness was still high after soaking, and the fabric sample strength also retained stability. Water had difficulty penetrating into the interlayers of fabric sample and adhering to the working surface. The cooling and lubricating effect of water helps reduce average underwater COF. Meanwhile, compared to the speeds and loads under dry sliding, the influence of speed on average underwater COF increased, and the influence of load on average underwater COF decreased. Generally, greater loads and higher speed causes a large actual force-bearing area and small local stress, reducing COF. Consequently, the COF in Figure 4 was lower than Figure 3.

### 3.2. The Effect of Epoxy Resin and Hybrid Fabric

In Figure 5a, the cross-section fiber is Nomex, and the longitudinal fiber is PTFE. In Figure 5b, the cross-section fiber is PTFE, and the longitudinal fiber is Nomex. Figure 5c shows the working surface before wear. The high adhesion of epoxy resin enhances the interlayer bonding force directly. Epoxy resin exhibits high strength and serves as a rigid phase to provide support to hybrid fabric. PTFE fibers serve as the soft phase to increase toughness. Nomex fibers mainly provide support for PTFE fibers and ensure strong adhesion to epoxy resin.

In Figure 5d–f, under dry sliding, the working surface becomes flatter after wear. Flat surface morphology helps reduce COF. PTFE fibers and epoxy resin also show great synergy [13]. PTFE fibers embedded in the epoxy resin matrix had a perfect lubricating effect, which is beneficial for reducing COF and improves wear resistance [14]. Slight abrasive wear occurs, and the abrasive particles have a second lubricating effect that can decrease COF further.

In Figure 5g–i, with respect to underwater lubrication, the high strength of epoxy resin and large bonding force interlayers remain stable; afterwards, water had difficulty penetrating between the interlayers and adhered to the surface to reduce COF. Thus, Nomex fibers absorb less water and retained strong adhesion to epoxy resin. As a result, the hardness of working surface was high after soaking and the fabric laminate sample maintained high strength underwater. Finally, the fabric laminate sample enhance its tribological properties underwater via the high strength and large bonding force of epoxy resin.

### 3.3. The Effect of Graphite Fillers

In Figure 6, the underwater wear amount and water absorption after soaking 24 h were all obtained by weighing using a precision balance (accuracy: 0.1 mg). The wear amount and the water absorption first decreased and then increased with graphite content increasing. When graphite fillers reached the content of 5 wt.%, water absorption and wear amount were the smallest. Therefore, 5 wt.% graphite contents underwater are the most suitable ratio [23]. During the curing process, a certain content of fillers was beneficial for reducing the internal stress generated in the resin. Adding fillers made the bubbles in the resin more likely to overflow. Thus, the resin matrix and fabric are more closely combined, enhancing mechanical strength and reducing water absorption of the fabric composite. Small water absorption results in little decreases in underwater surface hardness, resulting in smaller underwater wear amount [28]. Finally, an appropriate amount of graphite filler was beneficial for improving underwater tribological properties [33].

Graphite particles are mainly embedded in an epoxy resin matrix, as shown in Figure 7a,b. Figure 7c shows a single graphite particle embedded in resin matrix with strong adhesion. The graphite particles are exposed to the working surface during the friction test. Graphite is an excellent solid lubricant that is conducive for reducing COF. The graphite particles protruding on the surface of the epoxy resin also have good load-carrying capacity [24,25]. Finally, graphite filler increases wear resistance and decreases COF. Graphite also gathers in Nomex fiber bundles, as shown in Figure 7d; they reunite in the pits of the Nomex fiber bundles. Graphite has stable performance underwater. The combination of graphite filler, epoxy resin matrix and Nomex fibers showed strong underwater bonding force with high strength.

In Figure 7e, a dense and continuous underwater transfer film formed on the surface of metal pin. The transfer film will further improve wear resistance and reduce COF. In Figure 7f, graphite particles are transferred to the metal pin, participating in the formation of transfer film. The appropriate content of graphite fillers enhanced the quality of the transfer film on the metal pin [22].

### 3.4. Laster Microscope Observations of Working Surface

Figure 8 shows the 3D profile of the working surface; Sa is the surface roughness, The Sa of the working surface was 4.0 µm before wear, 2.8 µm after dry sliding wear and 1.8 µm after water-lubricated wear. v1 and v2 are the wear volume of 0.044 mm^2^ under dry sliding and 0.045 mm^2^ under water-lubricated. The working surface becomes flatter after wear with the lowest underwater surface roughness and high underwater wear resistance. The wear marks are shallow on the surface because of the high hardness and strength of epoxy resin.

In Figure 8b, under dry sliding, the generated abrasive particles affect surface roughness via movement with the rotational test fabric sample or by being crushed on the friction surface. Abrasive particles adhering to the friction surface could reduce wear amount under dry sliding. In Figure 8c, with respect to underwater lubrication, the generated abrasive particles are taken away by the lubricating water continuously, lowering surface roughness. Due to its high strength and excellent adhesion of epoxy resin, improvements in surface hardness and reductions in the water absorption of fabric samples were attained. Moreover, a reasonable content of graphite filler is conducive for forming the transfer film on the metal pin; as a result, wear resistance was high underwater.

### 3.5. SEM Images of the Worn Area under Water Lubrication after Soaking

Under water lubrication, in Figure 9a,b,e, PTFE fibers have slight adhesive wear. In Figure 9c,d, PTFE fibers have rolling deformation. Figure 9d is a partially enlarged view of Figure 9c, where rolling deformation is more serious. Adhesive wear and rolling deformation are all conducive to the outward diffusion of PTFE. The working surface is rich in PTFE fibers with adhesion and deformation, and the soft PTFE fibers can easily form a lubricating film on the hardness working surface and metal pin [34,35]. Ultimately, PTFE fibers have an excellent lubricating effect on the working surface.

In Figure 9g, Nomex fibers were firmly bound in epoxy resin before wear. Figure 9f is a partially enlarged view of the Nomex fibers in Figure 9e. Figure 9h is a partially enlarged view of Figure 9i. In Figure 9f,i, the Nomex fibers are rolled into a flat shape with a smoother surface after wear. The worn Nomex fibers are still firmly bound in the epoxy resin. There is no noticeable peeling of Nomex fibers after wear. It shows that Nomex fibers and epoxy resin are bonded together firmly under water-lubricated test after soaking.

## 4. Conclusions

This paper mainly describes the hybrid PTFE/Nomex fabric/epoxy resin graphite reinforced composite with enhancing underwater tribological properties. The conclusions are as follows:(1)The fabric sample has a lower underwater average COF and high underwater wear resistance. Underwater tribological properties were improved by the epoxy resin and graphite filler. The high strength and excellent adhesion of epoxy resin improved in water-wear resistance. The graphite filler and PTFE fibers reduced underwater COF.(2)The high strength of epoxy resin provides good support for hybrid fabrics. The underwater interlayers bonding force remains strong after soaking. The surface of Nomex fibers was flatter and smoother without noticeable peeling and was still firmly bound in the epoxy resin after wear.(3)Graphite filler acts as a solid lubricant for improving wear resistance and reducing COF via a good load carrying capacity and participating in the formation of transfer film on the metal pin. Adhesive wear and rolling deformation of PTFE fibers occurred, both of which were conducive to the formation of transfer film on the metal pin.

## Figures and Tables

**Figure 1 materials-15-00062-f001:**
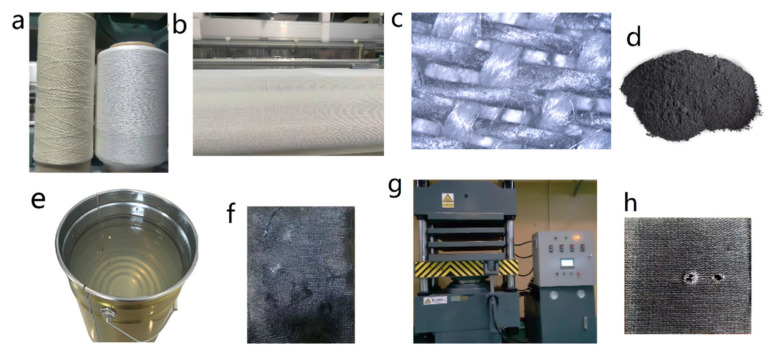
Fabric sample preparation process, (**a**) Nomex and PTFE fibers, (**b**) weaving machine, (**c**) microscope images of hybrid PTFE/Nomex fabric, (**d**) graphite filler, (**e**) epoxy resin, (**f**) pre-impregnated fabric, (**g**) curing press and (**h**) fabric sample.

**Figure 2 materials-15-00062-f002:**
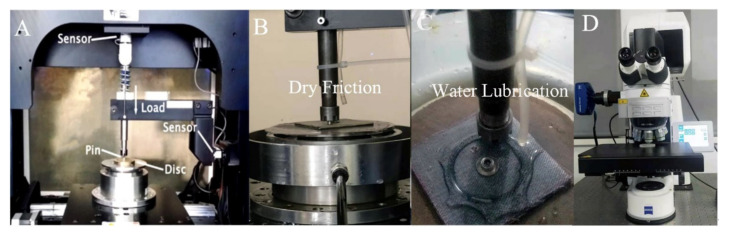
(**A**) The schematics of pin-on-disk tribometer, the photos of (**B**) friction test under dry friction, (**C**) friction test under water lubrication and (**D**) laser scanning microscope.

**Figure 3 materials-15-00062-f003:**
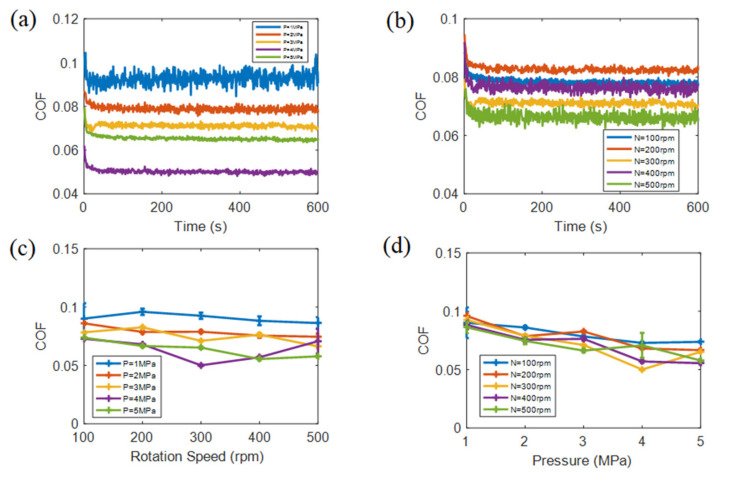
The COF under dry sliding, (**a**) COF vs. time with different loads under the middle speed of 300 rpm, (**b**) COF vs. time with different speeds under middle load of 3MPa, (**c**) average COF vs. speed and (**d**) average COF vs. load.

**Figure 4 materials-15-00062-f004:**
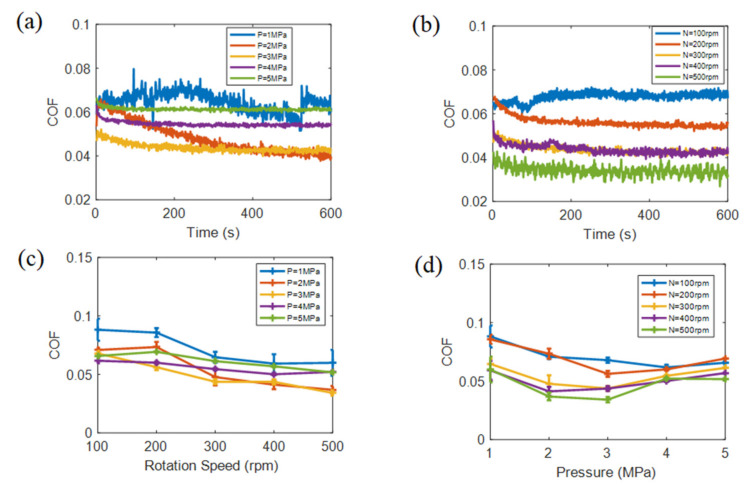
The COF under water lubrication, (**a**) COF vs. time with different loads under the middle speed of 300 rpm, (**b**) COF vs. time with different speeds under middle load of 3MPa, (**c**) average underwater COF vs. speed and (**d**) average underwater COF vs. load.

**Figure 5 materials-15-00062-f005:**
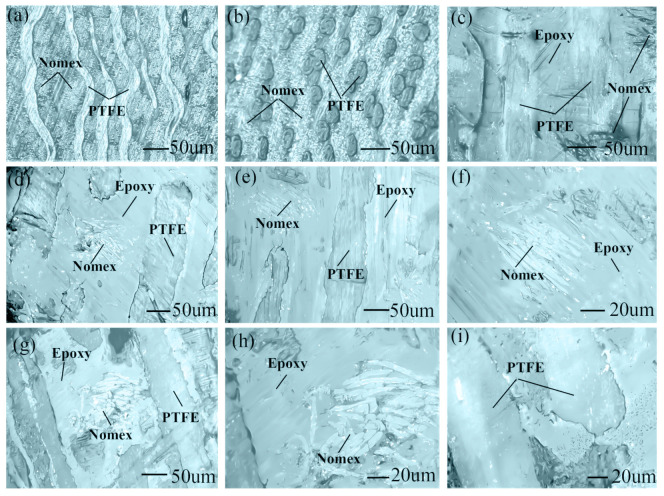
Microscope images of fabric laminate sample, (**a**) vertical cross-section, (**b**) horizontal cross-section, (**c**) working surface. (**d**,**e**) dry friction worn surface and (**f**) Nomex fibers after dry friction wear. (**g**) Water-lubricated wear surface, (**h**) Nomex fibers and (**i**) PTFE fibers after water-lubricated wear.

**Figure 6 materials-15-00062-f006:**
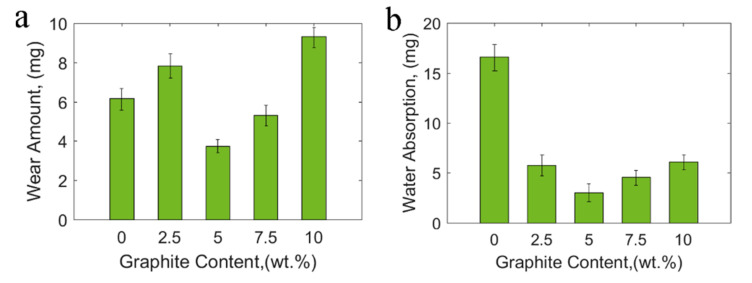
(**a**) Water-lubricated wear amount and (**b**) water absorption after soaking 24 h with different graphite content.

**Figure 7 materials-15-00062-f007:**
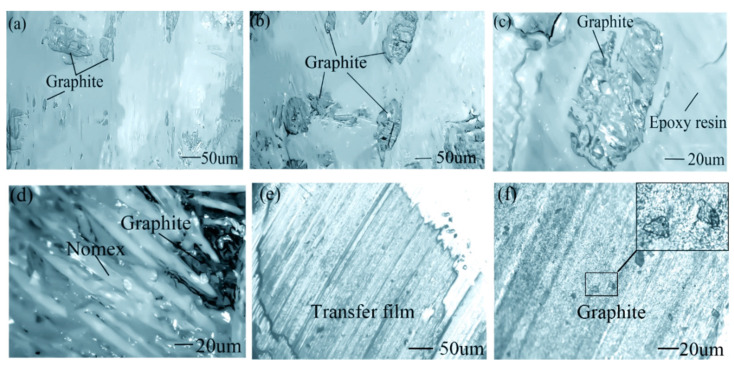
Graphite phase of the fabric laminate sample, (**a**,**b**) graphite embedded in the epoxy resin matrix, (**c**) an enlarged view of the graphite phase in (**b**) and (**d**) graphite embedded in Nomex fibers. (**e**) The transfer film on the pin and (**f**) the enlarge view of (**e**).

**Figure 8 materials-15-00062-f008:**
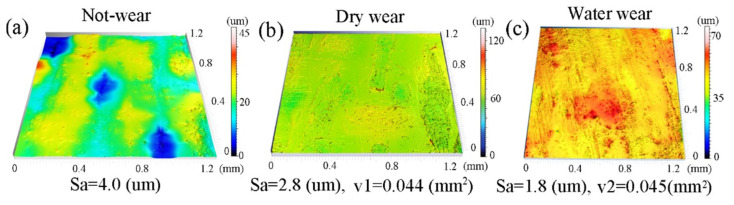
The 3D profile of the composite (**a**) before wear, (**b**) after dry sliding wear and (**c**) after water-lubricated wear.

**Figure 9 materials-15-00062-f009:**
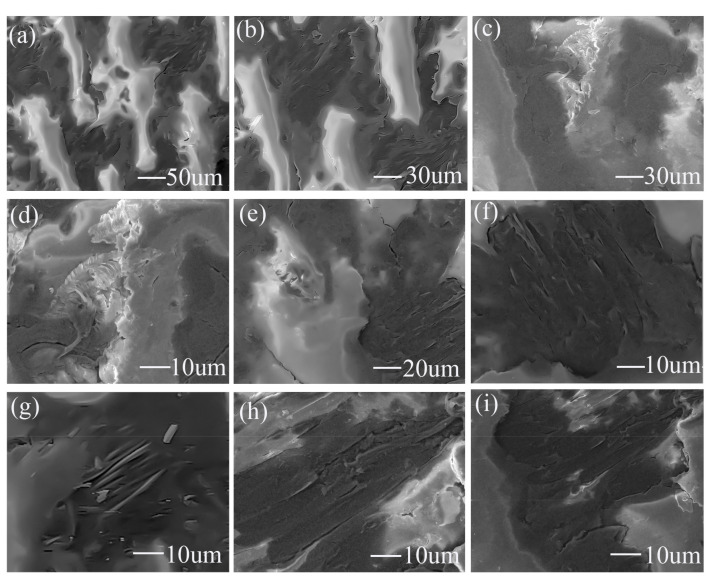
SEM images of the water-lubricated worn area after soaking, (**a**) surface morphology of the wear zone, (**b**) partially enlarged view of (**a**), (**c**) rolling wear zone of PTFE, (**d**) partially enlarged view of (**c**), (**e**) adhesive wear zone of PTFE, (**g**) Nomex fibers before wear, (**f**,**i**) Nomex fibers after wear and (**h**) partially enlarged view of (**i**).

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
