# Peer review of "Enhancement of the Water-Lubricated Tribological Properties of Hybrid PTFE/Nomex Fabric Laminate Composite via Epoxy Resin and Graphite Filler"

_materials, 2021, doi:10.3390/ma15010062_

Round 1

Reviewer 1 Report

The manuscript "Enhance the Tribological Properties of Multilayer Hybrid PTFE/Nomex Fabric Composite under Water Lubrication via Epoxy Resin and Graphite Filler" is well organised and written. The results and experimental work are presented in an appropriate manner. However, I would suggest that the some figures, such as Figure 3 and Figure 4 need proper attention, as the results presented are difficult to read and understand, keeping in view the importance of the figures for understanding the results. The figures must be enhnaced to better understand.

Reviewer 2 Report

Row 2   Enhance should be replaced by Enhancement.

Row 14 large should be replaced by high.

Row 273 Conclusion should be replaced by Conclusions.

Static friction coefficient was mentioned versus time. It is known that the static value of friction is detected only at the beginning of the motion, i. e. at zero time.   

Reviewer 3 Report

The paper is interesting and brings new information. It does suffer from some language issues. Also, the design and particularly the interpretation of the result of the wear testing could be improved.

Generally, "fiber" in singular is used, more appropriate would be to use "fibers" and change the text accordingly.

line 14: "because of" change to "connected to the"
l. 37: "to improve" -> "for improvement of"
l. 67: "Liu and Wang et al. [25]" -> "Liu et al. [25]"
l. 72/73: "by added CNTs, and the self-lubrication of CNTs was further graphitized" - reformulate, unclear
l. 77: "adding fillers are hard to enhance the adhesion between layers effectively" - what is hard? fillers? adding them? enhancing the adhension?
l. 84: "In this paper, more suitable epoxy resin was used under water lubrication." - repeats the same info as in line 34.
l. 114: "sample was fixed" -> "sample fixed"
l. 121-124: it would be more appropriate to provide the wear rate in m/s and length of the travel in meters, than only the duration of the test.
section 3:
The readablity of the figures is insufficient. The legends are too small. Now, the static friction coefficient is not very important, and also in my view is not very well definable from the graphs in Fig.3. Moreover, the plots in Fig. 3a and 3b look actually very similar, it does not seem to be significantly different. If the three tests were done at each condition, how were the averages calculated? The presentation of the plots in Fig.3c and 3d is unclear - what is the load in Fig. 3c, and why this is shown, and also what is speed in Fig. 3d, and why this one is shown? Probably the most important and relevant here are just plots 3e and 3f.
l. 158: "It seen" -> "It can be seen" 
l. 167: "remains" -> "retains"
Fig. 4 - again, poorly readable legends
l. 189: "remains" -> "retains"
l. 213: "bundle" -> "bundles"
l. 216: "And the graphite" -> "The graphite"
l. 217/218: repetition of the same idea
l. 240: "And abrasive" -> "Abrasive"
l. 243: "improving the surface soughness" -> "reducing/lowering the surface roughness"
The whole sentence in lines 244-246 is somewhat unclear. Consider reformulation.
Fig.8: The color coding is somewhat misleading. In Fig. 8c the whole picture seems to be off set - far from xero, that is why it is so yellow/red. It should be shifted towards zero, so that all three can be visually compared.
l. 279: "properties was improved" -> "properties were improved"
l. 281: "opposite support" - what does it mean?

Reviewer 4 Report

In this manuscript by Ying, et.al., the authors report on enhancing tribological properties of multilayer PTFE/Nomex under water lubrication. This work appears to be novel and can be recommended for publication after the authors address the following issues:

1. The main selling point of this work appears to be the good performance of this multilayer under water. However, there is not any sufficient discussion in the introduction section about its use case (i.e. at what condition does one need to use TENG underwater?).

2. As an extension to Q1, did the authors try using other solution for lubrication?

3. The introduction can be a little broader. For instance, can this technique be possibly applied to other types of TENG? such as  https://doi.org/10.1002/admt.201800166, and is there any other type of conductive materials besides graphite that can act as better filler? for instance, thermoelectric-TENG (https://doi.org/10.1016/j.mtphys.2021.100519)

4. Figure 3 and 4 discussion are merely describing the observation of the difference in COF, with little explanation about the mechanism of why they are different. The discussion can be expanded.

5. "When the graphite fillers were at a content of 5wt.%, the water absorption and wear amount was the smallest" , why is this so?

overall, while this work appears to be novel, and there are many interesting observation, not enough was done to explain the "why". The authors can improve the manuscript by focusing on the "why", and more robust conclusion can be reached.

Round 2

Reviewer 4 Report

Unfortunately, the authors have not taken serious effort to improve their paper based on my earlier suggestion. For instance, in Q1, the authors merely reemphasize what is already in the introduction. Same as question 2.
In addition, my main concern is in Q4 about figure 3 and 4, to which the authors answered:
"We deleted the static friction coefficient figures that had little effect on the results,"

why is there a need to delete? this sounds dubious and suspicious at best. Instead of taking the opportunity to improve their paper and demonstrate their understanding, the authors chose to avoid answering the problem by deleting?

I would therefore recommend rejection for this paper.
